# 'Giving birth on a beach': Women's experiences of using virtual reality in labour

**Lorna Massov** [1]*, **Brian Robinson**[1], **Edgar Rodriguez-Ramirez**[2], **Robyn Maude**[1]

**1** School of Nursing, Midwifery and Health Practice, Victoria University Wellington, Wellington, New Zealand,
**2** School of Design, Victoria University, Wellington, New Zealand

* lorna.massov@vuw.ac.nz

**Data Availability Statement:** All relevant data are within the manuscript.

**Funding:** The author(s) received no specific funding for this work.

## Abstract

### Introduction

Birth is a normal physiological process, and many women want a natural birth. Women use a range of non-pharmacological pain relief methods to reduce labour pain intensity, to help manage labour pain and to induce relaxation. The purpose of this study was to explore the experiences of women using Virtual Reality as a non-pharmacological method of pain relief in labour. Virtual Reality has been shown to be an effective distraction technique in other acute pain settings which also reduces anxiety.

### Methods

This study conducted qualitative in-depth interviews postnatally with women who used Virtual Reality in labour. Thematic analysis was used to analyse the qualitative data.

### Results

Nineteen women used Virtual Reality in labour. Results from interviews with nineteen women in the postnatal period identified three main themes: impact of virtual reality on experience of labour, managing the pain of labour and challenges of using virtual reality in labour.

### Conclusion

This study identified that Virtual Reality was effective as a relaxation technique and helped in pain management by the use of self-efficacy techniques. Women in this study also identified preferred virtual environments specifically to use during labour and birth. This study provides a unique and original contribution to the field of Virtual Reality in labour and birth. It also identifies Virtual Reality as an acceptable and positive experience in the management of anxiety and labour pain.

**Competing interests:** The authors have declared that no competing interests exist.

## Introduction

Giving birth is a momentous, life changing experience for women that impacts upon the woman, her family and community. Pain during labour is also one of the most severe forms of pain a woman is likely to experience [1, 2] and is highly variable and subjective [3].

Labour pain is more than solely physiological, it is a complex emotional and psychological experience producing feelings of accomplishment and fulfilment [4, 5]. Therefore, efficacy of pain relief is not necessarily associated with maternal satisfaction [6, 7]. Non-pharmacological pain relief can help labouring women relax and remain in control of their pain, allowing them to be responsive to their body and the physiological process of birth, increasing their self-confidence and satisfaction with the birth experience [8]. Non-pharmacological methods women use in labour range from position changes, heat packs, water immersion, massage, and acupressure, to the use of distraction techniques that include music therapy, visualisation, and hypnobirthing.

A novel non-pharmacological technique shown to be effective in the management of acute pain and anxiety in a diverse range of healthcare settings is virtual reality (VR) distraction therapy [9]. Virtual reality has elements of many natural therapies, such as distraction and visualisation with meditation and hypnotic features. Virtual reality is a technology which allows users to enter an immersive 3D computer-generated world using a headset described as creating experiences that "blur the line between reality and illusion, pushing the limits of our imagination" [10, pg.2]. Distinct from traditional forms of distraction such as listening to music or watching television, VR has emerged as an efficient and powerful distraction tool, generating multi-sensory inputs of sight, sound, touch, and more rarely taste and/or smell. It is this convergence of sensory input and interactivity that gives the user the illusion of entering a virtual world [11]. This illusion of presence and immersiveness are considered central to the therapeutic value and efficacy of VR as analgesia [12].

Virtual reality has historically been recognised for its entertainment value, for example, in the area of console games. It is now used in education, simulation training and healthcare settings due to the availability of affordable, portable and high-quality headsets and software [13]. It is used in a variety of healthcare settings focusing on pain reduction, anxiety and distress for paediatric and adult patients undergoing painful procedures including burn wound care, physical therapy, chemotherapy, dental procedures or routine procedures such as venepuncture and intravenous catheter placement [14–19].

The application of VR in labour and birth has been explored with a range of research studies. These include systematic reviews and meta-analyses, randomised controlled trials and recently qualitative research. A recent systematic review and meta-analysis of randomised controlled trials reviewed eight studies. The following outcomes could be included: pain, anxiety, and satisfaction scores as well as the duration of first and second stage labour. Pooled results from the homogenous studies showed that VR significantly reduced the Visual Analogue Scale (VAS) scores during labour. Virtual reality was shown to significantly reduce the anxiety score during labour. In terms of satisfaction, VR significantly improved the satisfaction score during normal labour compared to the control group. In terms of duration of first and second stage of labour, there was no significant difference between the intervention and no intervention group [20]. Several quantitative studies have published results examining the effects of VR on labour pain in women. A 2017 study conducted a randomised controlled trial of 60 primiparous parturient women during stages of labour, determined by cervical dilatation. Results found statistically significant lower pain scores in the intervention group, compared to the control group [21]. A more recent randomised controlled trial by Frey and colleagues [22], with 27 labouring women found statistically significant decreases in sensory, affective, and

cognitive pain and anxiety. This study also found that 82% of women reported using VR during labour and 70% were interested in new VR developments specifically for childbirth [22].

A recent qualitative study conducted in the Netherlands interviewed women on their VR experience during labour. All women were highly satisfied with VR use during labour and reported pain reduction during VR [23]. Another qualitative study of twenty patients noted a significant increase in VR's ability to improve their self-efficacy in managing pain while in labour. These patients described the experience of VR as allowing them to connect with breathing, helped them relax and distract them from labour pain intensity. Seventy percent of these patients believed that VR reduced their pain, 60% felt it reduced their anxiety and 100% would recommend VR to other labouring women [24]. This paper identified the need for more studies to understand the patient experience of VR. Our research study aims to address this gap in the literature in the area of patient experiences.

The majority of research in the area of VR and labour pain has been quantitative and relied on numerical scales to measure pain intensity. However, labour pain, like pain of any other kind, is not purely sensory but is a complex experience strongly affected by emotions, environment, memory, and motivation [25, 26]. While quantitative research can give evidence that an intervention works or not, it cannot tell us why. The assessment of labour pain in research should reflect its complexity and multidimensional nature, thus a qualitative approach was appropriate in achieving the research aim which was to explore the experiences of women using VR in labour.

## Methodology

This study used an exploratory sequential mixed methods research design with three phases. The first phase was a qualitative in-depth interview with 25 pregnant women in the antenatal period. The second phase was a within-subjects crossover intervention study with 14 of the 25 women who used VR in active labour. The third phase was qualitative in-depth interviews postnatally with 19 of the 25 women who used VR in early and active labour. The antenatal qualitative findings and the quantitative results are published elsewhere [27, 28]. The mixing of methods can strengthen a study, allowing us to understand phenomena more fully, to gain in-depth and broader insights due to the wider range of views and perspectives presented [29].

## Methods

This paper reports on the results of the third phase of the study, the qualitative in-depth interviews conducted in the postnatal period. Ethical approval was obtained from Victoria University of Wellington Human Ethics Committee (VUWHEC 0000027393). Recruitment for this study commenced on 10 March 2019 and was completed on 10 October 2019. Women in this study either gave birth at home or at the sole tertiary hospital maternity unit under the care of their Lead Maternity Carer (LMC) midwife. Eighteen women birthed at the tertiary hospital and 1 woman birthed at home. The VR headset that the women used in labour was 'Oculus Go' developed by Meta Reality Labs and a range of software was uploaded from this platform https://www.youtube.com/watch?v=4lCwRg9oQ1c, https://www.youtube.com/watch?v=-bIWI_5ZOEk, https://www.youtube.com/watch?v=Io9Ned8o-pU. Women accessed a range of relaxing VR environments when in labour. These included tropical beach scenes, underwater dolphin scenes and safari scenes with animals. Only one woman had used VR before. Women used the VR in labour in a variety of ways. Some women used it while in the birthing pool, some used it while sitting on the swiss ball while others just sat while using the VR. Women were given the VR headset antenatally and were taught how to use the headset and navigate the menu by the first author. In-depth interviews were conducted in the postnatal period with

women who used VR in their labour. The women were recruited via their Lead Maternity Carer (LMC) midwife, when attending antenatal appointments, through email or antenatal education classes. The women self-selected to be participants in this study. Inclusion criteria were either nulliparous or multiparous, 35 weeks and over pregnant and 18 years of age or over. Twelve women were primiparous and seven women were multiparous. Women were excluded if they had any contraindications to VR use. This would include women who had a history of seizures, vision or earing deficits, history of psychiatric disturbances and history of severe nausea or a predisposition to motion sickness. Women gave written consent to participate.

Nineteen women used the VR in their labour and were interviewed in the postnatal period between 1–2 weeks after birth. The first author scheduled and conducted all informant interviews in the women's homes. The first author is a midwife and acknowledges having prior knowledge and beliefs in regard to labour and birth. The interview schedule comprised a single open-ended question to commence the interview which was: Can you tell me about your experience of using virtual reality in your labour and birth as if I am your pregnant friend? This was followed up by secondary questions and probes as the interview unfolded. Each interview was digitally audio-recorded and transcribed by the first author. Participants received copies of the transcribed interviews, and these were checked for accuracy by the participants. Women were assigned numbers to protect their identity.

Thematic analysis using the six step approach based on Braun & Clarke's writings [30] was the method of analysis. The research question was exploratory and so suited an inductive approach, in which codes were driven by the data content on the basis of the participant's experiences. The first author and second author separately read a selection of transcripts and then jointly worked together to create initial coding schemes. Coding proceeded iteratively; related comments were grouped into themes. Further reading enabled themes to be clarified, with the sub-division and merging of themes until stable. The two authors independently reviewed data within each theme. Minor changes were made by consensus until stable themes were agreed on. Themes identified were scrutinised and reviewed in a recursive process, as a quality check and in an endeavour to capture the essence of the raw data.

## Results

### Characteristics of the participants

The demographics of the women who participated in the study are presented in Table 1. Women were not required to have had prior experience using VR before and their confidence in using technology was included in the demographic questionnaire.

The three main themes that were identified were: impact of virtual reality on experience of labour, with subthemes of: *virtual reality as a form of escapism*, *the relaxing and enjoyable experience of virtual reality*, the second theme, managing the pain of labour and the third theme, challenges of using virtual reality in labour.

### Impact of virtual reality on experience of labour

The theme impact of virtual reality on experience of labour describes the variety of ways in which VR had an impact on women's labour experiences. These included the effectiveness of VR as a distraction technique. Women identified VR as a form of escape from the medicalised hospital environment, from the boredom of a long, slow labour, and as a way of removing themselves from their present reality. Virtual reality also had an impact on their labour experience in terms of bringing enjoyment to the experience, helping to relax the women and reduce

**Table 1. Characteristics of women.**

| Characteristic | n (%) of total sample n = 19 |
|---|---|
| **Age (years)** | |
| Under 25 | |
| 25–34 years | 11 (57.8) |
| Over 34 | 8 (42.1) |
| **Ethnicity** | |
| New Zealand European | 11(57.8) |
| Māori | |
| Pasifika | |
| Other European | 5 (26.3) |
| Asian | 3 (15.7) |
| **Educational Qualifications** | |
| Secondary School 1–2 years | |
| Secondary School 3–4 years | |
| Tertiary | 19 (100) |
| **Primiparous** | 12 (63.1) |
| **Multiparous** | 7 (36.8) |
| **Comfortable with Technology** | |
| Not Very | 1 (5.2) |
| Very | 16 (84.2) |
| Extremely | 2 (10.5) |

anxiety while they were in labour. Finally, VR had an impact on their experience of birth, women perceived it as a positive experience, and it contributed to satisfaction with their birth.

## Virtual reality as a form of escapism

Women in this study described how using VR was a form of escapism from many aspects of their labour. Firstly, escapism from the pain they were experiencing in labour and from the relentless of time passing slowly while they were in labour. Women described how the VR aided them to escape from their immediate surroundings, often a medicalised birth room. Finally, women described how the VR enabled them to escape their actual reality, they could escape from the present. Virtual reality was an effective form of distraction and in many of the narratives, the word 'escape' was used to convey going to a different place, away from the pain of labour.

> ". . .it was really nice at the beginning, I remember wearing it in the early stages of labour, I was like, oh my God, I totally forgot about the contractions, it was great. . . I didn't even think about the pain, it was quite. . .absorbing. . .I just forgot about it [labour], I didn't pay attention to it, it distracted me to an extent. . .
>
> (Participant one)

Women referred to specific elements of the VR environment which helped to distract them. They identified aspects of the virtual environment (VE) which allowed them to focus on something other than the pain, these included images as well as sounds.

"I was focusing on the views, the different sounds and animals in the rainforest, I got distracted so I wasn't really focusing on how painful it was."

(Participant two)

". . .I could focus on what I was seeing and try and distract my mind and then every time the contractions were coming, I was focusing on the dolphins and things coming and it really helped me." (Participant three)

Women who were experiencing labour for the first time shared that they had been prepared for a long and slow labour. Using the VR in labour helped some of the women escape the boredom of their long labour and the relentless of the contractions.

"It was nice and relaxing and a distraction from, you know, I had been having contractions and the whole day was quite long already and it was just a nice escape."

(Participant four)

The escapism experienced, was such, that the women lost all concept of time and time appeared to pass more quickly. According to one woman, the impact of using the VR was that "*time wasn't time. . ..it moved a lot faster.*" Time was impacted not only in terms of the overall length of labour, but also regarding the time between contractions and how long the contractions lasted.

"When I was having a contraction, I think it just helped to distract me a little bit and it helped the time go quickly, the contractions almost seemed shorter than when I wasn't wearing the goggles."

(Participant five)

"It was good for me just to escape. . .so, distraction and the ability for me to go into my own little world and not have to look around me at the same clock that's going very slowly in the same medical room. . .

(Participant six)

The women enjoyed the distraction effect of the VR but also its ability to remove them from 'the sterile, medical room.' The women commented on the impersonal nature of the birthing rooms and how enjoyable it was that VR brought some colour and beauty into the environment. Immersing oneself in the VR, helped some women remove themselves figuratively from being in hospital.

". . ..it helped the feeling of being at the hospital, I think, which is quite a sterile place, it [the VR] made it more colourful and entertaining. . .it takes you to a different world, it makes your mind wander, to dream. . .Well, I can rest here and still be in this world of colour and fun and beauty."

(Participant seven)

For 10 women, the process of labour was associated with a need to withdraw or escape from the present. They described the juxtaposition that the labouring women often finds herself in, wanting support people around, yet also needing to go within themselves, to an internal place, which enabled them to cope and manage the intensity of the labour experience.

"I really like being present for all of it as well, but it was neat to just go somewhere else for a while."

(Participant five)

## The relaxing and enjoyable experience of virtual reality

All women found the VR intervention to be a novel and entertaining experience in their labour. They enjoyed using it and found that it helped them to relax, creating a sense of peace and calmness.

". . .I really loved the scene that I looked at, the beach scene. . . I was just sitting on the floor leaning against the chair and felt really calm and relaxed."

(Participant five)

For Jackie using VR was credited with being more effective in reducing anxiety than pain: "*totally reduced anxiety. . .more so than the pain.*" The enjoyment of the VR experience for several women was accentuated by the ability of the VR to invoke a happy connection or a special memory.

"The period when I was wearing the goggles was a very happy moment in the labour experience and I remember being very relaxed. . .it helped that the scene which was played on the VR featured manatees which looked remarkably like my dog and that I remember making me feel very happy."

(Participant eight)

Eighteen women felt that using the VR in their labour contributed to them having a positive birth experience and increased their satisfaction with the birth. For many, this positive birth experience was due to being relaxed and calm in labour and because it was novel and fun.

"It was good, it was fun, it was nice. Thinking back, I don't know what I would have done for such a long time, just waiting, and having pain if I didn't have the VR."

(Participant six)

"The memory of wearing the goggles is a very happy memory, so I feel lucky to have a good memory."

(Participant eight)

## Managing the pain of labour

The ability of the VR to assist the management of labour pain at different stages of the labour process was articulated by the women. The described the VR as helping them to manage their pain by controlling their breathing and giving them a sense of control over the labour. In one example, a moving image of a dolphin appearing, and disappearing was used as a reference point for the similar ebb and flow of contractions.

". . .I saw the dolphin was coming and the contraction was coming, I could focus, that is the contraction and then the dolphin was moving and then I was moving with that. . .so focus

on your object of reference, and the contractions are going to come and go like the dolphin, so it would appear and then it would go, so I think it was quite nice, very relaxing, I love it."

(Participant three)

"I feel it did [have an effect on relaxation], depending on what the scene was. The natural beach scene with the sound of the waves coming in and out assisted me to relax possibly by helping my breathing to be more regular."

(Participant nine)

As a way of managing labour pain, several features in the virtual environment were used by the women to assist in coping and tolerating the pain of labour. For example, participant nine synchronised her breathing with the visualisation and sound of the bubbles in the Dolphin VE. The combination of the sound and the image and the immersive experience intensified the effectiveness of the distraction.

"It has that in line with hypnobirthing without having to do all the hard work for hypno-birthing which is why I was keen to give it a go, being able to physically see an image rather than imagining an image is cool and . . .trying to capture all these senses, like the underwater one. I really loved that when you breathed the bubbles were there and so you heard to noise and saw the noise, so it's trying to capture these different senses to distract."

(Participant nine)

The idea of self-efficacy or a belief in one's ability to manage the pain of contractions was described by the women. One woman (participant seven) described her experience in labour and the impact of the VR as helping her to feel calm which gave her a sense of being in control: "I think by feeling calmer, I did feel more relaxed and in control for some of my labour." For other women the VR helped by giving them confidence that they could get through labour.

"I could finish a contraction thinking I'm not sure whether I could do this and then open my eyes and go, oh, there's an elephant, so it added a bit of positivity."

(Participant six)

Twelve women recalled feeling reductions in pain (sensory pain) and spending less time thinking about pain (cognitive dimension).

"So, using the VR in the first stage I felt that it took the edge of the pain, so I could focus on what I was seeing and distract my mind and then every time the contractions were coming, I was just trying to focus on the dolphins and things coming and it really helped me, so that my pain went from a 6 to 4. . .so I felt that it helped me."

(Participant three)

Using the VR in labour was a valued asset to help the women to relax, to focus on breathing techniques, which in turn helped them to manage the pain of the labour contractions.

"It helped with the breathing and the relaxation which then helped with the pain. It was nice to have my headphones with my meditation there and then the VR. It was having

options, staying in the right headspace. It was the right thing because then that was what helped with the pain, not getting stressed, not getting overwhelmed."

(Participant six)

## The challenges of using virtual reality in labour

Using VR in labour presented some unexpected challenges because of the hardware of the VR headset, and some surprises because of unexpected reactions such as the impact on the connection with others during labour. Despite these challenges most women shared something positive about the benefits of using VR in labour.

Sixteen women shared that their plans for managing their labour included using water immersion. Knowing that the VR headset was not waterproof was therefore a barrier to its effective use. Some women were anxious about damaging the equipment and the headset restricted their movements in the water.

"When I was in the water, I was concerned that I would get the VR goggles wet and then because I chose to be on all fours in the pool, the goggles, the weight in front was dropping my head. . .they were very heavy. . .when I tested them outside the water, it worked superfine. . ."

(Participant three)

The VR scenes were consumer ready products. Some of the scenes were designed were short periods of time of meditation and relaxation. Using these short scenes in labour was annoying for the women as they needed to be constantly replayed. Women described just getting into a VE, finding it relaxing and calming, and it just stopped. There were no scenes that had a running time of more than 10 minutes.

"So, the [VR] just ran out when I was starting to get absorbed in it and you've broken out of that zone and you're trying to find a different scene. . ..and there was one scene which had more people in it and I just didn't like that, I didn't like walking past all those random strangers."

(Participant nine)

Using the VR in other than a sitting position (which the VR headset is used in most cases) was uncomfortable and awkward for six women. The headset felt heavy, particularly in the neck area. This heaviness was problematic if a woman used it while they were on all fours, that is while they had both their knees and hands/arms on the floor. This position of being on all fours also proved difficult for the women when viewing aspects of the VR. Their vision was limited to just what was in front of them, they missed objects meant for distraction.

"Because I wanted to be on all fours, when I was having a contraction then I was just looking at sand."

(Participant five)

An aspect of the VR that some women mentioned that they did not enjoy when using it in labour was due to the immersive effect of VR. They shared that the all-encompassing aspect of

the goggles resulted in them feeling disconnected at times from their partner and from their labour experience.

> "I probably just wanted to be a bit more engaged with family...my partner and my midwife...I was going to have family come up briefly as well...it can be a good thing to take you away from your room and your scene, but I just wanted to make sure my partner was involved as well."

(Participant nine)

Despite these challenges of using VR in labour, 94% of women in this study would use VR in labour again and 94% would recommend VR to a pregnant friend.

## Discussion

This study explores the experiences of women using VR during labour. Our study supported findings from the only two qualitative studies published to date, demonstrating that VR is effective as a distraction technique in reducing anxiety, reducing labour pain intensity and improving women's self-efficacy [23, 31]. However, our study explored the effect of VR on birth experience. This is an area for future research and the effect of VR on fetal heart rate patterns and length of labour.

The main findings from our study demonstrated that VR had an impact on the experience of labour in a variety of ways. It was a useful intervention to manage labour pain, however VR proved challenging at times. These challenges ranged from the headset feeling heavy when using it in the birthing pool, or when using the VR in different positions. In a recent study on VR, women described VR as helpful during the first stages of dilation but when the contractions became too severe and they felt pressure or the urge to push they found the VR glasses too uncomfortable [23]. Other challenges identified in our study were women feeling removed from their support people/partners. However, only two out of the 19 women found this a challenge. As partners weren't interviewed for this study, we could not determine how they felt about their partner using the VR, this would be an interesting area for future research. In Muster & colleague'sstudy, they noted that overall partners were not bothered by their wife using VR [23]. Virtual reality was described as an effective distraction technique by most of the women, allowing them to shift their focus away from the pain of contractions. Similarly results from other qualitative studies found that VR was a good distraction method during labour and helped the participants to "focus on something else" while labouring [23, 31]. In our study the distraction of using the VR provided the women with a sense of being able to 'escape'. Women described how they wanted to escape during the worst of the pain. They used VR as a coping strategy. Wanting to escape from labour pain is not a new finding but knowing that VR is a useful tool to aid escaping this pain has not been established previously. A study explored women's perceptions of labour pain and the strategies they used to manage and prepare for the pain. Women prepared for and managed the pain of contractions with different strategies, by 'mental pain management' through utilising creative imagination, de-stressing prior to the birth, going to a 'happy place' and using distraction techniques like counting during a contraction [32]. Virtual reality is a useful tool to aid these strategies.

Descriptions of how the VR helped to manage their labour pain by allowing the women to relax and remain calm were conveyed in our study. Scenes in the VR were used to encourage breathing techniques, remain focused and to control anxiety. This was a similar finding to other qualitative studies. Patients in one study reported that the use of VR was perceived as calming and relaxing and helped them "connect with their breathing" [31]. In another study

women described VR as a relaxing experience, focusing on the nature environment, soothing voice and their breathing [23]. Studies on the use of relaxation techniques in labour found that it helped with creating a sense of calm and a way to manage pain, giving women the opportunity to rest between contractions, recuperate and restore their energy [8]. Being relaxed and calm in labour is also associated with an increased sense of control. The idea of control in labour is discussed extensively in birth literature with a recognised relationship between control and childbirth satisfaction [6, 33] and this is consistent with the experiences of the women in our study. The VR helped them to manage labour pain through distraction, helping them to relax, be calm and feel in control.

The need to withdraw from the 'present reality' was important for some women in our study, the VR allowed them to escape the feeling of surveillance. Literature on the ideal environment for birthing emphasises privacy, silence and low stimulation [34]. Ironically, the technology of the VR assisted the women to go within themselves and made them feel like they were going to another world. Virtual reality provided women with the ability to change their environment. Hospital birthing rooms disappointed the labouring women in our study, they described them as sterile, medicalised environments devoid of colour and warmth. The physical environment is described as unhelpful when it has a clinical 'hospital room' atmosphere with a lack of space and privacy [35]. In our study women described putting on the VR headset and being transported to a pleasant, colourful landscape, with relaxing surround sound, transfixed by underwater scenes, or roaming animals in the savannah, escaping to another world.

This study was exploratory in nature and while it was conducted with a relatively small sample size, this does not invalidate the data gathered from women's perspectives and experiences. Women in our study reported that they were highly motivated to use non-pharmacological methods of pain relief. In terms of demographics the women were mostly tertiary educated and identified as New Zealand European ethnicity. The postnatal interviews relied on recall up to 10 days after the women's birth experiences. However, in a literature review, previous studies have demonstrated that women's recollection of their labour and birth experience is surprisingly accurate years after the event [36]. Women self-selected to participate in this study thus the findings from this study cannot be generalised to the entire population of childbearing women, however this study offers a unique perspective and insight on the use of VR as a non-pharmacological method of pain relief in labour and birth. These results would be useful for all health professionals working in the area of maternity/obstetrics.

Women in this study identified a number of specific design recommendations for any future VE's specifically for use in labour. As all of the VE's the women viewed were consumer ready products they weren't designed for long periods. The VE's were too short—often only 10 minutes long. The women started to engage in the VE and then it stopped. A longer VE would possibly be more engaging and distracting for use in labour. Similarly women in Muster & colleague's study advocated for a prolonged VR session, this could have a better effect and longer lasting pain reduction in labour [23]. In a recent study participant's recommended less repetition and technical improvements in graphics [31]. Further research using a labour specific VE is required, with different lengths of content, emotionally engaging content for extended periods, identification of features that aid relaxation and reduce pain intensity during both early and active labour. For example, during active labour when pain is most intense, VE's with many distraction elements may be more useful. Further research should also be conducted to identify VR that is easy to use and intuitive for both women and midwives. Sixteen women wanted to use the VR in the water, the headset we used was not waterproof. There are waterproof headsets available. This would be the ideal environment for using VR, the combined analgesic effects of water immersion and VR and sitting relatively still in the water.

The responses from the women who participated in this study suggests that VR can be an acceptable and effective non-pharmacological method of pain management in labour helping women relax during labour. Studies in pain alleviation have shown that virtual reality is more effective than traditional forms of distraction, which is why this study utilised this intervention and also why this form of pain management technique should be explored further for use in labour. There are obvious challenges in using VR in labour, as this study identified. Women in our study, however, found it effective in managing labour pain through distraction, and by assisting them to relax and remain calm.

## Acknowledgments

The corresponding author wishes to thank all the co-authors for their assistance with this paper. The author wishes to acknowledge all the participants in the research. No grant was received to complete this project.

## Author Contributions

**Conceptualization:** Lorna Massov, Brian Robinson, Edgar Rodriguez-Ramirez.

**Data curation:** Lorna Massov, Edgar Rodriguez-Ramirez.

**Formal analysis:** Lorna Massov, Brian Robinson.

**Investigation:** Lorna Massov.

**Methodology:** Lorna Massov, Brian Robinson, Edgar Rodriguez-Ramirez, Robyn Maude.

**Supervision:** Brian Robinson, Edgar Rodriguez-Ramirez, Robyn Maude.

**Visualization:** Lorna Massov.

**Writing – original draft:** Lorna Massov.

**Writing – review & editing:** Lorna Massov, Brian Robinson.

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
