## [Decision Letter · Decision Letter 0]

15 Jan 2024

PONE-D-23-33730'Giving Birth on a Beach' Women's Experiences of Using Virtual Reality in LabourPLOS ONE

Dear Dr. Massov,

Thank you for submitting your manuscript to PLOS ONE. After careful consideration, we feel that it has merit but does not fully meet PLOS ONE’s publication criteria as it currently stands. Therefore, we invite you to submit a revised version of the manuscript that addresses the points raised during the review process.

We look forward to receiving your revised manuscript.

Kind regards,

Renato S. Melo, PhD

Academic Editor

PLOS ONE

3. Please could you confirm in the Methods section that recruitment for this study started on 10 March 2019 and was completed on 10 October 2019.

Additional Editor Comments:

Dear authors, the reviewers have evaluated the manuscript and decided that it has merit, however, it still needs some adjustments before it can be considered for publication. Therefore, I decided to reconsider the manuscript after major revisions.

Reviewers' comments:

Reviewer's Responses to Questions

**Comments to the Author**

1. Is the manuscript technically sound, and do the data support the conclusions?

Reviewer #1: Yes

Reviewer #2: Yes

2. Has the statistical analysis been performed appropriately and rigorously? 

Reviewer #1: I Don't Know

Reviewer #2: N/A

3. Have the authors made all data underlying the findings in their manuscript fully available?

Reviewer #1: Yes

Reviewer #2: Yes

4. Is the manuscript presented in an intelligible fashion and written in standard English?

Reviewer #1: Yes

Reviewer #2: Yes

5. Review Comments to the Author

Reviewer #1: TITLE: The title has giving birth on the beach , however through out the manuscript, there is no where the beach has been emphasized or specified as being the main place viewed by the women.

METHODS: In line 128 , the platform the authors are referring to is not clear. In line 135 , the authors should explain what they mean by self selected.

RESULTS: From the statements in lines 360-363, it seems some of the women delivered under or in water , if this is so it should be reflected in the results .

Reviewer #2: General comment:

I would like to congratulate the authors with the attempt to close the gap in previous research done on this subject. An intersting subject in the world of ever-changing technology and possibilities.

I do however have a number of comments, mainly with the intention of being more precise on how this study closes the gap intended, and how coming research should proceed from here.

A few comments on how to help the reader unfamiliar with the subject, to find further information.

Line 100-102: reference

Line 110-114: 25 included : describe 14 out of 25, 19 out of 25. The 19 out of 25 are described in the methods section, the 25 have been described elsewhere (citation needed) – the 14 out of the 25 needs explanation. This section needs clarification.

114-115: reference – where has it been published

127: Oculus Go: cite manufacturer, link to platform. Help the reader to find the software if interested.

138: possible contraindications to VR use needs clarification. Important for readers that are not familiar to VCR

Line 355: is it viable to use VCR in water or should this be discouraged even though they may be waterproof (line 473) : the problem seemed to be weight as well?

Line 356: are consumer ready products suitable for this specific function – relaxation during labour – or should the be more suitable products – this having been shown in previous studies (lines 466- 470) – can you be more specific into which research needs to be done apart from time ?

Any description of which scenes / music the included participants preferred or not preferred, in general – during the different stages of labour. For example: “I feel it did [have an effect on relaxation], depending on what the scene was” line 301

“For example, Liz synchronised her breathing with the visualisation and sound of the bubbles in the Dolphin” line 307

The term “many women” / “most women” are very general terms. Be more precise: how many (numbers or percentage)

Any differences between women giving birth for the first time and those having given birth previously?

The conclusion is very positive – 94 % of the participants would use it again ( these percentages have not been mentioned before), however challenges have been described. The challenges need to be discussed in the discussion as these challenges need to be addressed in further research

There are a number of well-chosen references in the introduction. How does the present study compare to previous studies: does the study support previous findings, does the present study close the gap intended or are further studies needs if yes – in which direction should these studies be directed, if not – why not

Are there any other aspects of being in labour that may influence the use of VCR?

Does VCR appeal to all women in labour? Table 1 in mind.

6. PLOS authors have the option to publish the peer review history of their article (what does this mean?). If published, this will include your full peer review and any attached files.

Reviewer #1: No

Reviewer #2: No

---

## [Author Response · Author response to Decision Letter 0]

5 Mar 2024

I have included all my specific reviewer comments in Response to Reviewer in supporting files.

---

## [Decision Letter · Decision Letter 1]

1 Apr 2024

PONE-D-23-33730R1'Giving Birth on a Beach' Women's Experiences of Using Virtual Reality in LabourPLOS ONE

Dear Dr. Massov,

Thank you for submitting your manuscript to PLOS ONE. After careful consideration, we feel that it has merit but does not fully meet PLOS ONE’s publication criteria as it currently stands. Therefore, we invite you to submit a revised version of the manuscript that addresses the points raised during the review process.

We look forward to receiving your revised manuscript.

Kind regards,

Renato S. Melo, PhD

Academic Editor

PLOS ONE

Journal Requirements:

Reviewers' comments:

Reviewer's Responses to Questions

**Comments to the Author**

1. If the authors have adequately addressed your comments raised in a previous round of review and you feel that this manuscript is now acceptable for publication, you may indicate that here to bypass the “Comments to the Author” section, enter your conflict of interest statement in the “Confidential to Editor” section, and submit your "Accept" recommendation.

Reviewer #1: (No Response)

Reviewer #2: All comments have been addressed

2. Is the manuscript technically sound, and do the data support the conclusions?

Reviewer #1: Yes

Reviewer #2: Yes

3. Has the statistical analysis been performed appropriately and rigorously? 

Reviewer #1: N/A

Reviewer #2: N/A

4. Have the authors made all data underlying the findings in their manuscript fully available?

Reviewer #1: Yes

Reviewer #2: Yes

5. Is the manuscript presented in an intelligible fashion and written in standard English?

Reviewer #1: No

Reviewer #2: Yes

6. Review Comments to the Author

Reviewer #1: INTRODUCTION: Line 96- it is not clear which gap the authors are referring to. The statement in lines 98-100 should be in the discussion section because the authors are commenting on their findings , also lines 101-102.

METHODS: For clarity, the authors should briefly explain how the virtual reality headset is or was used because form the text , it seem some women used them in some sort of pool (lines 368-369 and 371-374) while others sat done while using them (387-388). In lines 317 and 318 VE should 1st be written in full. In line 332 do the authors mean --- feel calm ---( instead of feel cam)?

Reviewer #2: Giving Birth on a Beach' Women's Experiences of Using Virtual Reality in Labour PONE-D-23-33730R1

I would like to congratulate the authors on their work in this interesting field which may be at contribution to the implementation and improvement of VCR in the future.

No further comments

7. PLOS authors have the option to publish the peer review history of their article (what does this mean?). If published, this will include your full peer review and any attached files.

Reviewer #1: No

Reviewer #2: No

---

## [Author Response · Author response to Decision Letter 1]

24 Apr 2024

Thank you to Reviewer #1 for their suggested changes - they have all been made.

---

## [Decision Letter · Decision Letter 2]

10 May 2024

'Giving Birth on a Beach' Women's Experiences of Using Virtual Reality in Labour

PONE-D-23-33730R2

Dear Dr. Massov,

We’re pleased to inform you that your manuscript has been judged scientifically suitable for publication and will be formally accepted for publication once it meets all outstanding technical requirements.

Kind regards,

Renato S. Melo, PhD

Academic Editor

PLOS ONE

Additional Editor Comments (optional):

Reviewers' comments:

Reviewer's Responses to Questions

**Comments to the Author**

1. If the authors have adequately addressed your comments raised in a previous round of review and you feel that this manuscript is now acceptable for publication, you may indicate that here to bypass the “Comments to the Author” section, enter your conflict of interest statement in the “Confidential to Editor” section, and submit your "Accept" recommendation.

Reviewer #1: All comments have been addressed

2. Is the manuscript technically sound, and do the data support the conclusions?

Reviewer #1: Yes

3. Has the statistical analysis been performed appropriately and rigorously? 

Reviewer #1: I Don't Know

4. Have the authors made all data underlying the findings in their manuscript fully available?

Reviewer #1: Yes

5. Is the manuscript presented in an intelligible fashion and written in standard English?

Reviewer #1: Yes

6. Review Comments to the Author

Reviewer #1: (No Response)

7. PLOS authors have the option to publish the peer review history of their article (what does this mean?). If published, this will include your full peer review and any attached files.

Reviewer #1: No

---

## [Editor Report · Acceptance letter]

22 May 2024

PONE-D-23-33730R2 

PLOS ONE

Dear Dr. Massov, 

I'm pleased to inform you that your manuscript has been deemed suitable for publication in PLOS ONE. Congratulations! Your manuscript is now being handed over to our production team.

Kind regards, 

on behalf of

Dr. Renato S. Melo 

Academic Editor

PLOS ONE